# Detecting LLM Memorization through Input Perturbation Analysis

## Abstract

Detecting memorization in LLMs is essential for assessing privacy risks, intellectual property exposure, and the reliability of benchmark evaluations. Yet existing detection methods are constrained by two practical limitations. They either require access to the training corpus to verify verbatim reproduction, or rely on logits that are sometimes unavailable in commercial black-box deployments. We introduce PEARL, a black-box framework that audit memorization essentially based on a model's input and output behavior. PEARL operationalizes the input perturbation sensitivity hypothesis (PSH): memorized instances occupy narrow attractors in input space and degrade sharply under semantically preserving perturbations, while generalized instances remain stable. Building on this principle, PEARL produces an instance-level memorization score through a calibrated comparison between the perturbation neighborhoods of known and unknown samples, requiring neither training data nor logit access. We evaluate PEARL on the Pythia model suite, with sizes ranging from 70M to 2.8B parameters, and find that its detection performance scales monotonically with model capacity, reaching AUC 0.81 on Pythia-2.8B and substantially outperforming both the gray-box ACR baseline (AUC $\approx 0.59$), the black-box CDD baseline (AUC $\approx 0.57$) and gray-box membership inference upper bound (AUC $\approx 0.67$). We further show that PEARL and membership inference attacks methods are complementary. They agree on fewer than half of their detections, and together identify 80.7% of true members, a 33% relative gain over the strongest individual method. This establishes PEARL as a practical, training-data-free auditing tool that captures generative memorization missed by existing approaches. Our anonymized code is available: `https://anonymous.4open.science/r/PEARL-497C/`.

## 1 Introduction

Large language models (LLMs) trained on web-scale corpora have repeatedly been shown to reproduce training data verbatim, including personally identifiable information, copyrighted text, and proprietary content (Carlini et al., 2021b; Hartmann et al., 2023b). This memorization scales with model size, data duplication, and prompt length (Carlini et al., 2023; Zhang et al., 2023; Schwarzschild et al., 2024), and now forms the technical basis of copyright litigation against major model providers (Grynbaum & Mac, 2023). It also confounds benchmark evaluations when test data leaks into training (Zhou et al., 2023), and exposes users to documented privacy risks (Yan et al., 2024; Lukas et al., 2023; Yao et al., 2024). Detecting memorization in deployed models is therefore a pressing concern, yet existing methods are poorly suited to the setting in which the concern is most acute. Most rely on prompting the model with exact prefixes drawn from the training corpus and checking whether the model produces the corresponding suffix (Carlini et al., 2023). This assumption actually fails where it is needed. Commercial models are accessed through black-box APIs, and their training data is undisclosed for strategic, legal, and security reasons (Zewe, 2024; European Parliament, 2023). Closing this gap requires detection methods that operate without the training corpus and without access to model internals.

**Relation to prior work.** Early memorization detection methods relied on white-box access, often identifying memorized content via high-confidence generations (Zhou et al., 2023). Black-box alternatives have since emerged. Carlini et al. (2023) use prefix prompting to detect exact suffix reproduction. Zhang et al.

(2023) formalize counterfactual memorization through prediction shifts when specific training instances are removed. More recent work analyzes output distributions across repeated prompts (Zhou et al., 2024), crafts detection prompts that elicit memorized content (Golchin & Surdeanu, 2024), and quantifies memorization through the minimal prompt length required to elicit it (Schwarzschild et al., 2024). These approaches share two limitations. First, they require either access to the training corpus or expensive retraining. Second, they collapse *memorization* into *membership inference attacks* (MIA). A datapoint that was seen during training is not necessarily memorized, since the model may have generalized from it rather than learned to reproduce it. We treat the two as operationally distinct, and our results in Section 6.7 confirm this distinction empirically by showing that membership inference and perturbation-based detection identify overlapping but non-equivalent populations of training instances. These limitations reduce to two practical questions. ❶ *How can memorization be detected when the training data is unavailable?* and ❷ *How can it be detected in a black-box setting without access to model internals?*

We address both questions with PEARL (**PE**rturbation **A**nalysis for **R**evealing **L**anguage model memorization), a black-box framework that requires neither the training corpus nor model weights or logits. PEARL rests on the *Input Perturbation Sensitivity Hypothesis* (PSH). If a model has memorized a datapoint, its output on that exact input is supported by a narrow attractor in representation space, and small semantically preserving perturbations will produce sharply different outputs. If instead the model has generalized, the same perturbations leave the output essentially unchanged. Brittleness under semantically preserving perturbations is therefore the behavioral signature of memorization, and it is observable from input and output alone. PEARL operationalizes this principle through a calibration step that compares the perturbation-stability distribution of known-member instances against that of known non-members, and uses the resulting gap to assign a per-instance memorization score together with a calibrated detection threshold. The fundamental idea that memorization manifests as brittleness is not novel, as it connects to several established research threads. The adversarial examples literature Szegedy et al. (2013); Goodfellow et al. (2014) demonstrated that models can be highly sensitive to input perturbations, though this work focused on security vulnerabilities rather than memorization detection. The novelty of PEARL lies in its specific operationalization of memorization detection through several key innovations, establishing brittleness as a primary detection mechanism rather than merely a correlative observation. This is achieved through a formal calibration procedure that systematically operationalizes the concept.

A particularly significant contribution is the framework's instance-level diagnostic capability. Unlike conventional robustness measures that provide model-level assessments, such as reporting aggregate robust accuracy, PEARL generates fine-grained, instance-specific memorization scores. This granularity enables precise identification of which specific examples a model has memorized, moving beyond broad characterization to targeted analysis. The calibration methodology itself represents a structured innovation through its use of proxy distributions, statistical gap measurement, and neighborhood-based decision rules. By comparing known training data against out-of-sample data to establish a baseline memorization gap $\gamma$ (defined below), and implementing a decision rule based on neighborhood aggregation, PEARL creates a reproducible detection system. Furthermore, the framework operates entirely at inference time, eliminating the computationally prohibitive retraining requirement of approaches like Feldman & Zhang (2020b), thus making memorization detection practical for contemporary large-scale models.

**Contributions.** We make the following contributions:

1. We introduce PEARL, a black-box framework for detecting instance-level memorization in LLMs that requires neither access to the training corpus nor model weights or logits.

2. We show that PEARL surpasses several baselines including gray-box MIA that require logit access. PEARL achieves AUC 0.805 on Pythia-2.8B, exceeding the gray-box MIA upper bound (AUC $\approx$ 0.67).

3. We demonstrate that memorization and MIA are empirically distinct and complementary signals. PEARL and MIA agree on fewer than half their detections (Jaccard $< 0.48$). Together they cover 81.1% of true members which represents a 33% relative gain over the strongest individual method while confirming that generative memorization constitutes a distinct privacy risk not captured by existing auditing tools.

4. We provide a systematic empirical evaluation across model scales, training epochs, perturbation operators, aggregation strategies, neighborhood geometries, and task domains (text completion and code synthesis), establishing the conditions under which PSH holds and characterizing its failure modes.

## 2 Problem formulation

### 2.1 Conceptual Foundation

The phenomenon of *memorization* in machine learning represents a fundamental departure from generalization. While commonly understood as a model "storing" specific data instances with minimal compression (Feldman, 2019), existing definitions often lack operational rigor for empirical detection. Our work builds upon established concepts in learning theory, particularly the principle of **uniform stability** (Bousquet & Elisseeff, 2002), which holds that generalization requires model outputs to be stable to the removal of any single training example. This connects directly to memorization via **leave-one-out analysis** (Feldman & Zhang, 2020a), which formalizes memorization through output sensitivity to training set composition. The core intuition behind PEARL is that memorized instances exhibit brittleness, lacking robustness to input perturbations which is a phenomenon related to findings in generalization studies (Zhang et al., 2017). Finally, we draw inspiration from **extraction-based definitions** (Carlini et al., 2021a), which demonstrate memorization through verbatim reproduction of training data, providing an empirical grounding for our framework. Building upon these foundations, we begin with a conceptual definition that captures the core theoretical understanding of memorization.

**Definition 1** (Memorization - Conceptual). *A model $\mathcal{M}$ has memorized a training instance $(x, y)$ when its performance on $x$ is critically dependent on $(x, y)$ being present in the training set, and the learned function exhibits negligible compression or generalization of the underlying data distribution at $(x, y)$ (Shalev-Shwartz & Ben-David, 2014; Feldman, 2019).*

While Definition 1 captures the essential nature of memorization, it is difficult to operationalize for empirical research. Verifying it requires counterfactual reasoning about training set composition, which in turn requires retraining the model with $(x, y)$ removed. At the scale of modern LLMs, this is computationally infeasible, and in deployment settings where the training set is undisclosed it is impossible.

### 2.2 The Input Perturbation Sensitivity Hypothesis (PSH)

To bridge the gap between the conceptual definition and a procedure that can be run at inference time, we propose the following hypothesis.

> ***Input Perturbation Sensitivity Hypothesis (*PSH*):*** *For a given model $\mathcal{M}$, task $t$, and data point $(x, y)$, if $\mathcal{M}$ has memorized $(x, y)$, then its performance on $t$ at $x$ will be sensitive to semantically-preserving perturbations of it, exhibiting greater degradation than would be observed for an instance the model has generalized.*

The key intuition is that memorized instances occupy a narrow attractor in the model's input space. The consequence of this is that the model can faithfully reproduce the training output for the exact original input, but nearby inputs produce divergent outputs because the model has no general representation to fall back on. Generalized instances, by contrast, occupy a smooth, broad basin where nearby inputs produce similar outputs because the model has learned a compact representation of the underlying pattern. This brittleness generalization duality connects to several established research threads. The adversarial examples literature Szegedy et al. (2013) demonstrated that neural models can be highly sensitive to input perturbations; PEARL repurposes this sensitivity as a diagnostic signal rather than an attack surface. More broadly, stability theory Bousquet & Elisseeff (2002) holds that properly learned functions should exhibit output stability to minor input variations—a criterion that memorized instances systematically violate. Brittleness under semantically preserving perturbations is therefore the behavioral signature of memorization, and crucially it is observable from input and output alone. Crucially, PEARL produces this signal at the instance level, yielding a per-example memorization score rather than an aggregate robustness statistic.

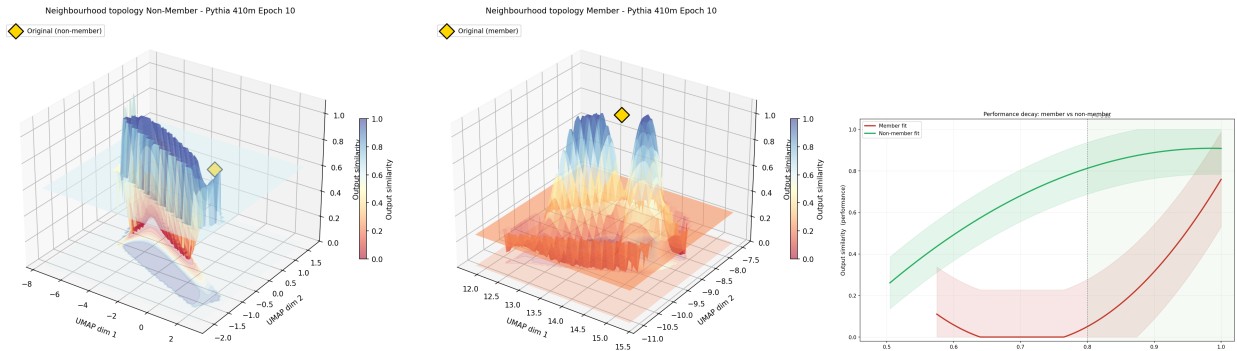

Figure 1: **Neighborhood topology of a non-member (left) and a member (center) instance** on Pythia-410M at epoch 10. The horizontal axes are the first two UMAP dimensions of the perturbation embedding; the vertical axis and color map encode output similarity $m_t(x'; \mathcal{M})$. The yellow diamond marks the original instance. *Left.* the non-member surface is smooth and uniformly high (blue), confirming stable generalization across its neighborhood illustrating the fact that the model's outputs vary little under perturbation (*broad basin*). *Center.* the member surface is jagged, with sharp peaks at the original input surrounded by near-zero output similarity valleys, consistent with a (*narrow attractor*) and memorization. *Right.* performance decay curves for members vs. non-members by considering the cosine similarity to quantity the similarity between the original and the modified input (in x-axis) and the similarity between the different generated output and the reference (in y-axis).

Figure 1 illustrates this geometric contrast at the instance level on Pythia-410M. The output-similarity surface of a non-member instance is smooth across its perturbation neighborhood, whereas that of a member instance is jagged, with sharp peaks at the exact input surrounded by near-zero valleys. This is the topological signature that PEARL detects.

### 2.3 Operational Definition

Building on the PSH, we introduce an operational definition that translates the conceptual understanding of memorization into a quantity measurable at inference time.

**Definition 2** (Memorization - PEARL Operational)**.** *For a task $t$ and a performance measure $m_t$, a model $\mathcal{M}$ has* memorized *an instance $(x, y)$ if its performance under $m_t$ is brittle: it exhibits significant performance degradation under semantically-preserving perturbations in its neighborhood $N(x)$, relative to the typical behaviour on non-memorized instances.*

**Remark 1.** *This definition complements stability-based approaches Feldman & Zhang (2020a) while operating purely at inference time, with no retraining required. The choice of $m_t$ and the construction of $N(x)$ are framework parameters that depend on the task; we instantiate both for next-token prediction in Section 3.*

**Validity condition.** The operational definition is sound only when the neighborhood $N(x)$ satisfies the **semantic preservation** requirement. Each perturbations $x' \in N(x)$ must preserve the semantic content of $x$, so that any observed performance degradation is attributable to memorization-induced brittleness and not to the model receiving meaningfully different input. Section 5 describes how we operationalize this requirement through a cosine similarity threshold in embedding space and empirically validate it in Section 6.6.

## 3 Methodology

PEARL operates on five components. We denote by $\mathcal{M}$ a model trained on a dataset $\mathcal{D}_{\text{train}}$, by $t$ a task such as next-token prediction or classification, and by $m_t(\mathcal{M}(x)) \in \mathbb{R}$ a task-specific performance measure that scores the model's output on an input $x$. For each instance $x$, we further introduce a neighborhood function $N(x)$ that returns a set of semantically preserving perturbations of $x$, and an aggregation operator $\mathcal{A}$ that

reduces a set of scalar performance scores to a single value. Together these components yield a per-instance brittleness signal.

**Definition 3** (Neighborhood Aggregation Score). *The* neighborhood aggregation score *for a data point* $(x, y)$ *under* $\mathcal{M}$ *is defined as:*

$$S(x, y) = \mathbf{A}\left(\{m_t(x', y'; \mathcal{M}) \mid (x', y') \in N(x, y)\}\right)$$

A high value of $S(x, y)$ indicates that the model's performance is largely preserved across the neighborhood, consistent with a generalized representation. A low value indicates significant degradation under perturbation, consistent with the brittleness signature of memorization predicted by the PSH.

The choice of $\mathbf{A}$ determines which aspect of the neighborhood distribution the score reflects, and yields a family of PEARL variants:

- $\mathbf{A}_{\mathrm{mean}}$ (arithmetic mean) summarizes average robustness across the neighborhood.

- $\mathbf{A}_{\mathrm{min}}$ (minimum) summarizes worst-case robustness, highlighting the most disruptive perturbation.

- $\mathbf{A}_{\mathrm{var}}$ (negative variance) summarizes output consistency, rewarding instances whose outputs vary little across perturbations.

- $\mathbf{A}_{\mathrm{quantile}}$ ($\alpha$-quantile) provides a robust compromise between average and worst-case behavior, parameterized by $\alpha \in (0, 1)$.

Section 4.1 describes how a per-instance score $S(x, y)$ is converted into a memorization decision through calibration on reference distributions. Section 5 specifies the concrete instantiation of $m_t$ and $N(x)$ used in our experiments.

## 4 Calibration and Detection

The neighborhood aggregation score $S(x, y)$ introduced in Section 3 measures the brittleness of a single instance, but its absolute value carries no calibrated meaning. A score of 0.45 is neither high nor low in isolation. PEARL therefore proceeds in two phases. A *calibration* phase establishes a reference scale by computing $S$ on two contrasting sets of instances, and a *detection* phase applies a calibrated threshold to flag previously unseen instances as memorized.

### 4.1 Calibration Phase

Calibration relies on two reference sets drawn from the same data distribution. The *memorization-suspect set* $\mathcal{E} = \{(x_i, y_i) \in \mathcal{D}_{\mathrm{train}}\}$ contains instances known to have been seen during training. The *generalization baseline set* $\mathcal{G} = \{(x_i, y_i) \notin \mathcal{D}_{\mathrm{train}}\}$ contains instances known to lie outside the training set.

Membership in $\mathcal{D}_{\mathrm{train}}$ is sufficient but not necessary for memorization (Feldman & Zhang, 2020a), so $\mathcal{E}$ may contain both memorized and properly generalized instances. This conservative composition dilutes the measured gap $\gamma$ (Definition 4) and makes the resulting decision rule stricter. A cleaner $\mathcal{E}$ could be obtained by retaining only instances the model can reproduce verbatim (Carlini et al., 2021a), at the cost of an additional filtering step. The baseline set $\mathcal{G}$ must be drawn from the same distribution as $\mathcal{D}_{\mathrm{train}}$, since otherwise any observed brittleness gap would conflate memorization with out-of-distribution effects. Section 5.2 describes how $\mathcal{E}$ and $\mathcal{G}$ are constructed in our experiments. The formulation of instances as pairs $(x_i, y_i)$ follows standard machine learning convention where $y_i$ represents the target output for evaluation. For LLMs, this accommodates various task-specific framings. In next-token prediction, $y_i$ would be the subsequent token(s); in instruction following, $y_i$ would be the expected response; and in sequence completion, $y_i$ could be the continuation of $x_i$. This pairing ensures PEARL can measure performance degradation relative to expected behavior across diverse tasks.

From these sets, we compute reference score distributions:

$$\mathcal{S}_{\mathcal{E}} \; = \; \{\, S(x,y) \,|\, (x,y) \in \mathcal{E} \,\}, \qquad \mathcal{S}_{\mathcal{G}} \; = \; \{\, S(x,y) \,|\, (x,y) \in \mathcal{G} \,\}.$$

**Definition 4** (The Memorization Gap)**.** *The* memorization gap *of PEARL under aggregation operator* $\mathcal{A}$ *is*

$$\gamma \; = \; A(\mathcal{S}_{\mathcal{G}}) - A(\mathcal{S}_{\mathcal{E}}).$$

The PSH predicts $\gamma > 0$, since memorized instances should exhibit lower neighborhood aggregation scores than generalized ones (Bousquet & Elisseeff, 2002; Feldman & Zhang, 2020a). We verify this prediction empirically in Section 6.4.

**Remark 2.** *While various aggregation operators A can be used to define the gap, the mean ($\mu$) is a common choice that facilitates intuitive interpretation and enables the use of Z-score formulations for standardized effect size measurement.*

For more nuanced analysis, we define a *memorization score* as:

$$M_{\text{score}}(u) = \frac{\mu(S_{\mathcal{G}}) - S(u)}{\sigma(S_{\mathcal{G}})}$$

where $\sigma(\cdot)$ denotes standard deviation. The score $M_{\text{score}}(u)$ quantifies how anomalously brittle an instance is relative to the generalization baseline (Arjovsky et al., 2020), providing a continuous signal that complements the binary decision rule introduced next.

### 4.2 Detection Phase

Given an unseen instance $u = (x_u, y_u)$, PEARL computes its neighborhood aggregation score $S(u)$ and applies a threshold calibrated against the reference distributions.

**Definition 5** (PEARL Decision Rule)**.** *PEARL flags u as memorized if:*

$$S(u) \; < \; A(\mathcal{S}_{\mathcal{G}}) \, - \, \tau\gamma,$$

*where $\tau \geq 0$ is a sensitivity parameter. With $A = \mu$ and $\tau = 1$, the rule reduces to $S(u) < \mu(\mathcal{S}_{\mathcal{E}})$. By the next of this study, the optimal value of $\tau$ is defined dynamically through Youden's optimal threshold (Schisterman et al., 2008)*

$$\tau = \frac{Youden \; Optimal \; Threshold}{\gamma}$$

The rule uses known membership labels exclusively for calibration, never for detection itself. The detection step depends only on the calibrated reference distributions $\mathcal{S}_{\mathcal{E}}$ and $\mathcal{S}_{\mathcal{G}}$, so the flag assigned to $u$ is determined by its own behavior under perturbation rather than by any prior knowledge of its training-set status. The sensitivity parameter $\tau$ allows the user to trade precision against recall by tightening or loosening the threshold around the calibrated gap. Algorithm 1 summarizes the full procedure.

## 5 Experiment setup

We evaluate PEARL through eight experiments. The first seven (RQ1 through RQ7) study detection quality, neighborhood design, capacity scaling, training dynamics, aggregation operator sensitivity, neighbourhood geometry, and the relationship to MIA, all on a text completion task. The eighth (RQ8) examines whether the findings transfer to a structurally different task, code synthesis. The full set of research questions is as follows.

- **RQ1 (Detection Quality).** How effectively does PEARL identify memorized instances compared with existing memorization and contamination detection methods?

---
**Algorithm 1** PEARL Calibration and Detection

---
1:  **procedure** PEARL-CALIBRATE($\mathcal{M}, \mathcal{E}, \mathcal{G}, N, A$)
2:      $\mathcal{S}_\mathcal{E} \leftarrow \emptyset, \quad \mathcal{S}_\mathcal{G} \leftarrow \emptyset$
3:      **for** $(x, y) \in \mathcal{E} \cup \mathcal{G}$ **do**
4:          scores $\leftarrow \{ m_t(x', y; \mathcal{M}) \,|\, x' \in N(x) \}$
5:          $S \leftarrow A(\text{scores})$
6:          **if** $(x, y) \in \mathcal{E}$ **then**
7:              append $S$ to $\mathcal{S}_\mathcal{E}$
8:          **else**
9:              append $S$ to $\mathcal{S}_\mathcal{G}$
10:         **end if**
11:     **end for**
12:     $\gamma \leftarrow A(\mathcal{S}_\mathcal{G}) - A(\mathcal{S}_\mathcal{E})$
13:     **return** $A(\mathcal{S}_\mathcal{E}), \ A(\mathcal{S}_\mathcal{G}), \ \gamma$
14: **end procedure**

15: **procedure** PEARL-DETECT($u, A(\mathcal{S}_\mathcal{G}), \gamma, \tau$)
16:     $S_u \leftarrow A(\{ m_t(x', y_u; \mathcal{M}) \,|\, x' \in N(x_u) \})$
17:     **if** $S_u < A(\mathcal{S}_\mathcal{G}) - \tau\gamma$ **then**
18:         **return** memorized
19:     **else**
20:         **return** generalized
21:     **end if**
22: **end procedure**

---

- **RQ2 (Transformation Efficiency).** How does the choice of neighborhood transformation affect the memorization gap $\gamma$ and detection reliability?

- **RQ3 (Impact of Model Size).** How does model capacity affect PEARL's detection performance, and how many instances does it identify across model sizes?

- **RQ4 (Effect of Repeated Exposure).** How does the $\gamma$ evolve as a function of the number of training epochs?

- **RQ5 (Aggregation Operator Sensitivity).** How does the choice of aggregation operator $A$ affect $\gamma$ and detection reliability?

- **RQ6 (Neighborhood Size and Similarity Threshold).** How does the number of perturbations $K$ per sample and the input cosine similarity threshold $\alpha$ affect detection reliability?

- **RQ7 (Memorization vs Membership).** To what extent do instances flagged as memorized by PEARL overlap with those identified as members by standard MIA methods, and what does each approach detect exclusively?

- **RQ8 (Task Dependency).** How does the sensitivity of PEARL and the different MIA methods vary following the task considered (text completion vs code synthesis) ?

## 5.1 Model

We evaluate PEARL on the Pythia family of open-source autoregressive language models Biderman et al. (2023), spanning four sizes (70M, 410M, 1.4B, and 2.8B parameters). We use checkpoints trained on the deduplicated variant of The Pile to control for memorization due to repeated training sequences while ensuring that any detected brittleness reflects genuine memorization rather than simple repetition from duplicated data. Two further properties of the Pythia suite make it well-suited to our setup. The availability of intermediate training checkpoints enables controlled analysis of how memorization accumulates during fine-tuning,

which we exploit in RQ4. The fully open-source nature of the suite with known training corpus, architecture, and public weights allows us to construct ground-truth membership labels, vary scale systematically, and compare against gray-box baselines under identical conditions.

## 5.2 Dataset

We evaluate PEARL on two task domains.

**Text completion (RQ1 through RQ7).** Training members $\mathcal{D}_m$ and non-members $\mathcal{D}_{\neg m}$ are drawn from the Mimir benchmark (Duan et al., 2024) and filtered with a 13-gram overlap threshold of 0.8 to suppress near-duplicate contamination. Each split contains 1,000 instances of mixed-domain text spanning prose, code, and structured data.

**Code synthesis (RQ8).** Members are drawn from CodeContest (Li et al., 2022), a corpus of competitive programming problems and solutions on which we fine-tune the model. Non-members are drawn from OpenCodeInstruct (Ahmad et al., 2025), an instruction-tuning dataset for code generation. Each split contains 500 instances.

## 5.3 Performance Measure and Neighborhood function.

**Instantiation.** For both task domains, we instantiate the performance measure $m_t$ and the semantic-preservation proxy (transformation) through the same embedding model, OpenAI `text-embedding-3-small`. Let $\phi(s)$ denote the embedding of a string $s$. The performance measure $m_t(x'; \mathcal{M})$ is the cosine similarity between the model's continuation on input $x'$ and the reference continuation $y$. The proximity function used to validate the neighborhood is the cosine similarity between perturbed and original inputs, $\cos(\phi(x'), \phi(x))$. Using the same embedding for both roles ensures that the validity criterion on $N(x)$ and the brittleness signal on $m_t$ live in a comparable representation space.

**Similarity threshold.** We set the default similarity threshold to $\alpha = 0.90$, retaining only perturbations $x' \in N(x)$ with $\cos(\phi(x'), \phi(x)) \geq 0.90$. This default is justified empirically in Section 6.6.

**Neighborhood size.** We generate up to $K = 5$ perturbations per instance, all within the $[0.90, 1.00]$ similarity bucket. Section 6.6 shows that $K = 5$ is sufficient to saturate the detection signal, with negligible variance across repeated draws.

**Transformation operators.** We construct $N(x)$ using five perturbation operators grouped into two families (Table 1). *Surface-level* operators make shallow syntactic edits that preserve token content and consistently produce perturbations within the $\alpha \geq 0.90$ bucket. *Advanced* operators alter lexical content more aggressively and, as Section 6.2 will show, often fail to produce neighbors that satisfy the validity criterion under the default configuration.

Table 1: Transformation operators used to construct $N(x)$.

| Family | Operator | Description |
|---|---|---|
| Surface | `whitespace` | Insert or remove whitespace characters |
| | `change_case` | Mutate character casing at random positions |
| | `swap_chars` | Swap adjacent characters at random positions |
| Advanced | `synonyms` | Replace tokens with synonyms |
| | `bitflip` | Apply bit-level character corruption |

Table 2: Detection performance of PEARL and baselines on Pythia-1.4B (1 000 members / 1 000 non-members from Mimir). AUC is the best value across 10 epochs. $\#\hat{M}_E$ / $\#\hat{M}_G$ = instances flagged as memorized in the member / non-member split at the best epoch.

| Method | Access | Best AUC | $\#\hat{M}_E$ | $\#\hat{M}_G$ |
|---|---|---|---|---|
| **Logit access required** | | | | |
| Loss Yeom et al. (2018) | Gray-box | 0.686 | 516 | 226 |
| Min-K% Shi et al. (2023) | Gray-box | 0.685 | 525 | 248 |
| Neighborhood Attack Mattern et al. (2023) | Gray-box | **0.687** | 585 | 284 |
| ACR Schwarzschild et al. (2024) | Gray-box | 0.586 | 320 | 142 |
| **Require prior knowledge of training data** | | | | |
| N-Gram ($n = 5$) Carlini et al. (2021a) | Black-box | **0.970** | 889 | 23 |
| N-Gram ($n = 7$) Carlini et al. (2021a) | Black-box | 0.968 | 504 | 93 |
| N-Gram ($n = 13$) Carlini et al. (2021a) | Black-box | 0.957 | 360 | 35 |
| **No access to training data / logits** | | | | |
| CDD Dong et al. (2024) | Black-box | 0.565 | 188 | 126 |
| PEARL ($A_{\mathrm{mean}}$) | Black-box | **0.777** | 592 | 144 |
| PEARL ($A_{\mathrm{median}}$) | Black-box | 0.740 | 633 | 218 |
| PEARL ($A_{q_{0.25}}$) | Black-box | 0.704 | 504 | 125 |
| PEARL ($A_{q_{0.10}}$) | Black-box | 0.674 | 441 | 120 |
| PEARL ($A_{\mathrm{min}}$) | Black-box | 0.622 | 538 | 281 |
| PEARL ($A_{\mathrm{neg\text{-}var}}$) | Black-box | 0.517 | 273 | 240 |

## 6 Results

### 6.1 RQ1: Detection Quality

**Setup.** We compare PEARL against two families of baselines, and n-gram overlap as a training-data-dependent reference. **CDD** (Dong et al., 2024) is a black-box memorization detector that measures the peakedness of the model's sampled-completion distribution. It requires no logit access and is PEARL's primary direct competitor. **ACR** (Schwarzschild et al., 2024) is a gray-box approach that flags memorization when a shorter adversarial prompt suffices to elicit the target string, requiring gradient access. **n-gram overlap** (Carlini et al., 2021b) declares a sample memorized when the model's continuation shares at least one $n$-word sequence with the reference training text, for $n \in \{5, 7, 13\}$. It requires access to the training corpus. We additionally include three MIA scores derived from token log-probabilities, namely Loss (Yeom et al., 2018), Min-K% (Shi et al., 2023), and Neighborhood Attack (Mattern et al., 2023). All methods are evaluated on Pythia-1.4B fine-tuned for 10 epochs on 1,000 members from Mimir, scored against 1,000 non-members, and we report the best AUC across 10 epochs. For PEARL, the flagged counts are taken at the calibrated detection threshold from Section 5.3. Table 2 reports the AUC for each method, together with the number of instances flagged as memorized in the member ($\mathcal{E}$) and non-member ($\mathcal{G}$).

**PEARL surpasses baseline black-box and gray-box methods.** On Pythia-1.4B, PEARL ($A_{\mathrm{mean}}$) achieves AUC 0.777 with a 13% relative gain over the best MIA method (Neighborhood Attack, 0.687) and a 37% relative gain over CDD (0.565), while requiring no access to the training data or the logits It also outperforms ACR Schwarzschild et al. (2024), which requires gradient access.

We observe, however, that due to the access to the training data, n-gram performs better, reaching AUC 0.970 on Pythia-1.4B. This is driven by the model's tendency to generate long verbatim continuations at late epochs. However, this method is only applicable when the training corpus is available for reference text matching, which is the setting PEARL is designed to circumvent. $n$-gram overlap and PEARL therefore address different operational scenarios and should not be considered direct competitors. We include this for completion and can be regarded as the maximum value that can be obtained when there is a perfect oracle (training data).

Table 3: Per-transformation performance on Pythia-1.4B at epoch 10 ($N_E = N_G = 1,000$). *Input sim.* is the mean cosine similarity between original and perturbed input while *Coverage* is the fraction of samples with at least one valid neighbor. AUC is computed per transformation independently.

| Group | Transformation | Input sim. | Coverage | $\mu(S_\mathcal{G})$ | $\mu(S_\mathcal{E})$ | $\|\gamma\|$ | AUC |
|-------|----------------|-----------|----------|------------|------------|--------|-----|
| Surface | `whitespace` | 0.975 | 100% | 0.763 | 0.472 | 0.290 | 0.789 |
| | `change_case` | 0.966 | 99.9% | 0.785 | 0.512 | 0.273 | 0.743 |
| | `swap_chars` | 0.957 | 100% | 0.805 | 0.522 | **0.283** | **0.798** |
| Advanced | `synonyms` | 0.747 | 72% | 0.416 | 0.316 | 0.100 | 0.615 |
| | `bitflip` | 0.619 | 79% | 0.330 | 0.273 | 0.057 | 0.549 |
| Surface (combined) | | — | 100% | 0.513 | 0.424 | **0.089** | **0.614** |
| Advanced (combined) | | — | 73% | 0.270 | 0.256 | 0.014 | 0.521 |

> **Finding (RQ1).** PEARL with $\mathcal{A}_\text{mean}$ aggregator outperforms all gray-box MIA methods, ACR, and CDD baselines while not having access to neither logit nor gradient access. These results confirm the fact that the model exhibits strong, exploitable perturbation sensitivity for memorized instances.

### 6.2 RQ2: Transformation Efficiency

**Setup.** We evaluate the five perturbation operators introduced in Section 5.3. **Surface-level** transformations (`change_case`, `whitespace`, `swap_chars`) make shallow syntactic edits that preserve token identity and consistently produce perturbations within the $\alpha \geq 0.90$ similarity range. **Advanced** transformations (*synonyms*, *bitflip*) alter lexical content more aggressively, producing neighbors at substantially lower mean similarity (0.747 and 0.619 respectively). We evaluate each operator independently and the two groups in combined configurations. All results are on Pythia-1.4B at epoch 10.

**Surface transformations produce consistently strong and valid neighbors.** All three surface operators yield AUC between 0.743 and 0.798 and memorization gaps between 0.273 and 0.290. `swap_chars` performs best in isolation (AUC 0.798, $|\gamma| = 0.283$), while the combined surface configuration reaches AUC 0.614, the highest of any single-level configuration, confirming the complementarity of minor syntactic perturbations. All three surface operators achieve mean input similarity at or above 0.957, well within the $\alpha \geq 0.90$ validity criterion established in Section 6.6.

**Advanced transformations produce invalid neighbors, not weak ones.** `synonyms` and `bitflip` yield near-chance AUC (0.615 and 0.549) and gaps below 0.1. This result must be interpreted carefully. It does not imply that lexical substitutions are uninformative about memorization. The mean similarity of these operators (0.747 and 0.619) places the majority of their generated neighbors outside the valid neighborhood $N_{0.90}(x)$. Within this invalid range, the observed performance drop conflates two effects, memorization-induced brittleness and the natural degradation any model undergoes when given a meaningfully different input. The near-chance AUC therefore indicates that advanced transformations, as currently implemented, do not generate semantically-preserving neighbors under the $\alpha = 0.90$ criterion, rather than that they carry no memorization signal. Additionally, `bitflip` achieves only 79% coverage, meaning 21% of instances receive no valid neighbor at all, further limiting its utility.

> **Finding (RQ2).** Surface-level transformations produces semantically-preserving neighbors and achieves higher AUC than Advanced transformation which generates neighbors of outside the threshold criterion.

### 6.3 RQ3: Impact of Model Size

**Setup.** We evaluate four Pythia variants (70M, 410M, 1.4B, and 2.8B parameters) at epoch 10 using PEARL ($A_\text{mean}$), with three gray-box MIA methods as upper-bound references. Detection counts are

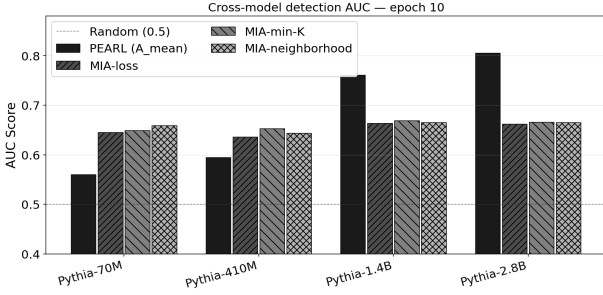 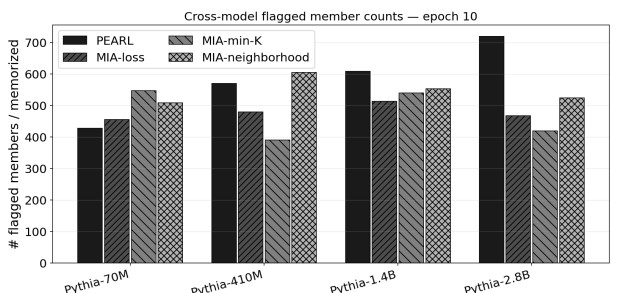

Figure 2: AUC at epoch 10 for PEARL ($A_{\mathrm{mean}}$) and MIA methods across model sizes. PEARL scales sharply with capacity while MIA remains flat ($\approx 0.65$).

Figure 3: Number of flagged members at epoch 10 (Youden threshold). PEARL flags up to $720/1\,000$ for Pythia-2.8B, nearly double the 428 for Pythia-70M.

reported at the Youden's J optimal threshold (TPR − FPR) on a balanced set of 1,000 members and 1,000 non-members. Figures 2 and 3 report the results.

**Larger models both memorize more and are more detectable.** As shown in Figure 2, AUC rises from 0.560 at 70M to 0.595 at 410M then jumps sharply to 0.761 at 1.4B and 0.805 at 2.8B. The memorization gap $|\gamma|$ follows the same pattern, growing from 0.051 to 0.071 to 0.213 to 0.254 (Figure 4). This non-linear improvement suggests a capacity threshold in the range [410M, 1.4B] beyond which models commit sufficiently to individual training instances for the perturbation-based signal to become strongly discriminative. Below this threshold, memorization exists but is shallower and harder to detect through output stability alone.

We also observe that at the Youden-optimal threshold, Pythia-2.8B flags 720 out of 1,000 members as memorized while maintaining fewer false positives than smaller models (Figure 3). Precision improves from 56.0% at 70M to 80.5% at 2.8B, indicating that the signal-to-noise ratio of PEARL's score grows with scale. This is consistent with the broader memorization literature Carlini et al. (2023); Tirumala

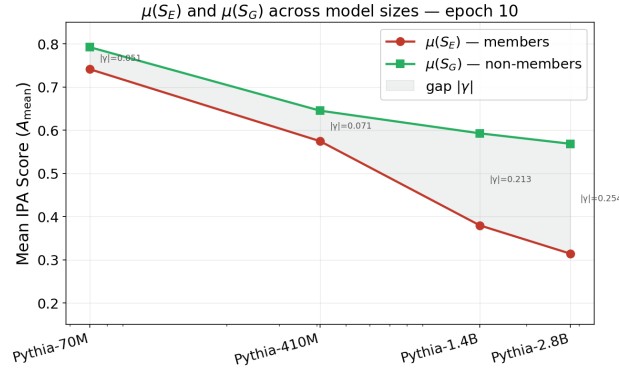

Figure 4: Evolution of $\mu(S_E)$ and $\mu(S_G)$ with model capacity at epoch 10 ($A_{\mathrm{mean}}$). The shaded area represents the memorization gap $|\gamma|$, which grows from 0.051 (70M) to 0.254 (2.8B).

et al. (2022), which shows that larger models have greater capacity to store training instances verbatim rather than compress them into generalizable representations.

**MIA performance does not improve with model scale.** MIA AUC ranges from 0.648 to 0.668 across all four models, a near-flat trajectory showing that log-probability signals saturate early and are insensitive to the additional memorization larger models exhibit. PEARL surpasses MIA AUC at 1.4B (0.761 vs. 0.661) and extends this advantage substantially at 2.8B (0.805 vs. 0.668), making it the only evaluated method whose detection quality scales with model capacity.

**Finding (RQ3).** PEARL is the only evaluated method whose detection quality scales with model capacity. MIA performance remains flat at AUC around 0.65. PEARL surpasses the gray-box MIA upper bound above 410M parameters, flagging up to 720 of 1,000 true members at 2.8B with 73.9% precision, without requiring any logit access.

### 6.4 RQ4: Effect of Repeated Exposure

**Setup.** Using the 11 Pythia-1.4B checkpoints (epochs 0 through 10), we track the evolution of $|\gamma|$ and AUC for all six aggregation operators. The results are shown in Table **??** and Figure 5. We observe a common two-phase pattern across all aggregation operators.

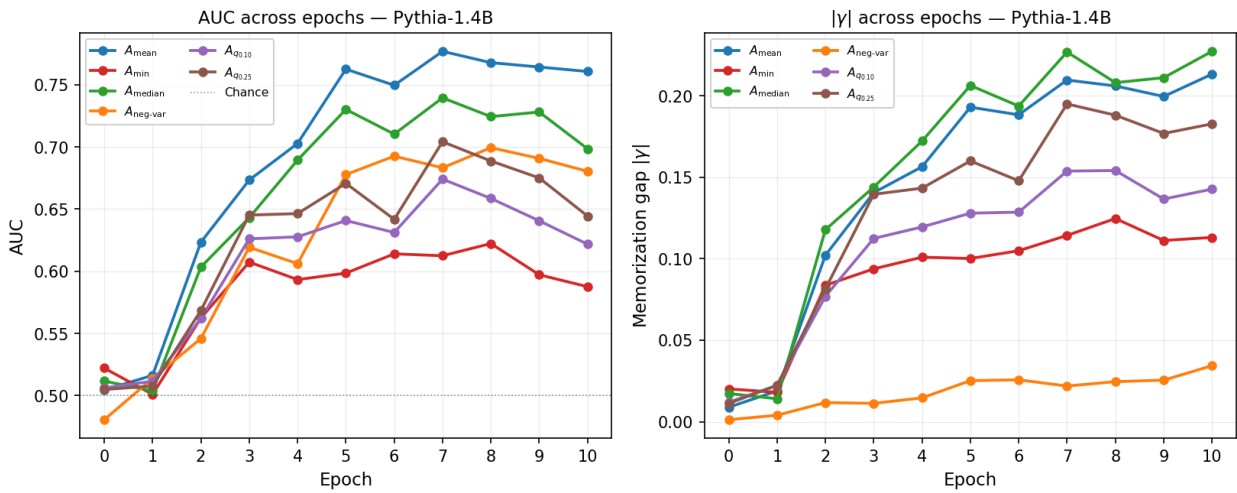

Figure 5: AUC and memorization gap $|\gamma|$ (right) across training epochs for all six PEARL aggregation operators on Pythia-1.4B.

Table 4: Evolution of $\mu(S_{\mathcal{G}})$ and $\mu(S_{\mathcal{E}})$ across training epochs on Pythia-1.4B (1,000 members / 1,000 non-members).

| Metric | Ep. 0 | Ep. 1 | Ep. 2 | Ep. 3 | Ep. 4 | Ep. 5 | Ep. 6 | Ep. 7 | Ep. 8 | Ep. 9 | Ep. 10 |
|---|---|---|---|---|---|---|---|---|---|---|---|
| $\mu(S_{\mathcal{G}})$ | 0.468 | 0.474 | 0.510 | 0.522 | 0.587 | 0.608 | 0.605 | 0.624 | 0.621 | 0.602 | 0.621 |
| $\mu(S_{\mathcal{E}})$ | 0.459 | 0.455 | 0.408 | 0.381 | 0.431 | 0.415 | 0.417 | 0.414 | 0.415 | 0.403 | 0.407 |

**Phase 1. Initial dormancy (epochs 0–1).** The first fine-tuning pass produces only a small signal: $|\gamma_{\text{mean}}|$ rises from 0.009 to 0.019 and AUC from 0.505 to 0.516. Both $\mu(S_{\mathcal{G}})$ and $\mu(S_{\mathcal{E}})$ increase marginally ($\Delta \approx 0.006$–0.007), suggesting the model is still redistributing internal representations from its pre-trained state without yet committing strongly to individual training instances. Notably, $\gamma_{\text{neg-var}}$ is positive from epoch 1 onward: the consistency signal aligns with the brittleness signal from the very first epoch, unlike smaller models where the variance signal lags behind for several epochs.

**Phase 2. Sharp acquisition (epoch 2).** At epoch 2, the memorization gap surges: $|\gamma_{\text{mean}}|$ jumps from 0.019 to 0.102 (5.4× increase) and AUC rises from 0.516 to 0.623 in a single epoch. $\mu(S_{\mathcal{E}})$ drops sharply (0.455 → 0.408) while $\mu(S_{\mathcal{G}})$ rises (0.474 → 0.510), widening the gap from both sides simultaneously.

**Phase 3. Sustained plateau (epochs 3–10).** From epoch 3, both $|\gamma|$ and AUC grow steadily, plateauing in the ranges [0.188, 0.213] and [0.750, 0.777] respectively. Peak AUC of 0.777 is reached at epoch 7. The

Table 5: AUC (top) and memorization gap $|\gamma|$ (bottom) per aggregation operator across training epochs (Pythia-1.4B, 1 000 members / 1 000 non-members). Bold = best value per epoch.

| Epoch | $A_{\mathrm{mean}}$ | $A_{\min}$ | $A_{\text{neg-var}}$ | $A_{q_{0.10}}$ | $A_{q_{0.25}}$ | $A_{\mathrm{median}}$ |
|---|---|---|---|---|---|---|
| | | | *AUC* | | | |
| 0 | 0.505 | **0.522** | 0.517 | 0.506 | 0.505 | 0.512 |
| 1 | **0.516** | 0.501 | **0.517** | 0.512 | 0.508 | 0.504 |
| 2 | **0.623** | 0.562 | 0.547 | 0.562 | 0.568 | 0.603 |
| 3 | **0.673** | 0.607 | 0.623 | 0.626 | 0.645 | 0.643 |
| 4 | **0.703** | 0.593 | 0.610 | 0.628 | 0.646 | 0.689 |
| 5 | **0.763** | 0.599 | 0.678 | 0.641 | 0.671 | 0.730 |
| 6 | **0.750** | 0.614 | 0.693 | 0.631 | 0.642 | 0.710 |
| 7 | **0.777** | 0.613 | 0.687 | 0.674 | 0.704 | 0.740 |
| 8 | **0.768** | 0.622 | 0.702 | 0.659 | 0.689 | 0.724 |
| 9 | **0.764** | 0.597 | 0.693 | 0.641 | 0.675 | 0.728 |
| 10 | **0.761** | 0.588 | 0.681 | 0.622 | 0.644 | 0.698 |
| **Best** | **0.777** | 0.622 | 0.702 | 0.674 | 0.704 | 0.740 |
| | | *Absolute memorization gap $|\gamma| = |\mu(S_E) - \mu(S_G)|$* | | | | |
| 0 | 0.009 | **0.020** | 0.001 | 0.012 | 0.011 | 0.017 |
| 1 | 0.019 | 0.018 | 0.006 | **0.022** | **0.022** | 0.014 |
| 2 | 0.102 | 0.084 | 0.013 | 0.077 | 0.082 | **0.118** |
| 3 | 0.141 | 0.094 | 0.013 | 0.112 | 0.140 | **0.144** |
| 4 | 0.156 | 0.101 | 0.016 | 0.120 | 0.143 | **0.172** |
| 5 | 0.193 | 0.100 | 0.028 | 0.128 | 0.160 | **0.206** |
| 6 | 0.188 | 0.105 | 0.029 | 0.129 | 0.148 | **0.194** |
| 7 | 0.210 | 0.114 | 0.024 | 0.154 | 0.195 | **0.227** |
| 8 | 0.206 | 0.125 | 0.027 | 0.154 | 0.188 | **0.208** |
| 9 | 0.200 | 0.111 | 0.029 | 0.137 | 0.177 | **0.211** |
| 10 | 0.213 | 0.113 | 0.039 | 0.143 | 0.183 | **0.227** |

plateau is stable through epoch 10 with no systematic decay, confirming that memorization patterns are durable once consolidated. The signal on 1.4B peaks slightly earlier and stabilises at a much higher level, indicating more decisive memorization commitment at larger scale.

> **Finding (RQ4).** On Pythia-1.4B, PEARL reveals a three-phase trajectory. It start with an initial dormancy period (epochs 0–1, AUC $\approx 0.51$) following by an acquisition jump at epoch 2 ($|\gamma|$ rises from 0.019 to 0.102, AUC from 0.516 to 0.623), and finally a sustained plateau through epochs 3–10 (AUC 0.75–0.78). The $A_{\text{neg-var}}$ consistency signal is positive from epoch 1, earlier than at smaller scales, reflecting faster entropy suppression for memorized instances in larger models.

### 6.5 RQ5: Aggregation Operator Sensitivity

**Setup.** We evaluate all six operators $A_{\mathrm{mean}}$, $A_{\min}$, $A_{\text{neg-var}}$, $A_{q_{0.10}}$, $A_{q_{0.25}}$, and $A_{\mathrm{median}}$ on Pythia-1.4B while keeping the neighbourhood function fixed at the combined surface-level combined configuration ($\tau = 0.90$, $K = 5$). Table 5 reports AUC and $|\gamma|$ at each epoch checkpoint for Pythia-1.4B.

$A_{\mathrm{mean}}$ **is the best aggregation operator followed by** $A_{\mathrm{median}}$. $A_{\mathrm{mean}}$ achieves the highest peak AUC (0.777 at epoch 7) and leads every epoch from epoch 2 onward. It provides a stable, interpretable signal and enables the Z-score formulation of $M_{\mathrm{score}}$ from Section 4.1. We also observe that $A_{\mathrm{median}}$ is the strongest robust alternative. $A_{\mathrm{median}}$ achieves peak AUC 0.740. At 1.4B, the memorization signal is strong enough that the central tendency of the distribution is highly informative, and the median's resistance to outlier neighbours makes it particularly stable. For applications requiring robustness to a small number of poorly-generated perturbations, $A_{\mathrm{median}}$ is the preferred alternative. While $A_{\mathrm{mean}}$ leads on AUC, $A_{\mathrm{median}}$ achieves the highest $|\gamma|$ at most epochs. The two operators are therefore complementary: $A_{\mathrm{mean}}$ maximizes discrim-

Table 6: Effect of the input cosine similarity threshold $\alpha$ on Pythia-1.4B at epoch 10. Only perturbations with input similarity $\geq \alpha$ are used for the $A_{\mathrm{mean}}$ score. Coverage = fraction of samples with at least one valid neighbor above the threshold.

| $\alpha$ | Coverage | $\mu(S_{\mathcal{G}})$ | $\mu(S_{\mathcal{E}})$ | $|\gamma|$ | AUC |
|---|---|---|---|---|---|
| $\geq 0.50$ | 100% | 0.621 | 0.407 | 0.213 | 0.761 |
| $\geq 0.60$ | 100% | 0.645 | 0.422 | 0.223 | 0.773 |
| $\geq 0.70$ | 100% | 0.684 | 0.445 | 0.239 | 0.790 |
| $\geq 0.80$ | 100% | 0.729 | 0.468 | 0.261 | 0.811 |
| $\geq 0.90$ | 100% | 0.794 | 0.508 | **0.286** | **0.836** |

Table 7: AUC as a function of the number of perturbations per sample $K$ on Pythia-1.4B at epoch 10 ($A_{\mathrm{mean}}$, 30 random trials). *Targeted*: only [0.90, 1.00] bucket. *Pooled*: all five buckets.

| $K$ | Targeted ($\alpha \geq 0.90$) | | Pooled ($\alpha \geq 0.50$) | |
|---|---|---|---|---|
| | AUC (mean) | AUC (std) | AUC (mean) | AUC (std) |
| 1 | 0.759 | 0.019 | 0.677 | 0.036 |
| 2 | 0.794 | 0.011 | — | — |
| 3 | 0.814 | 0.007 | 0.728 | 0.030 |
| 4 | 0.826 | 0.004 | — | — |
| 5 | **0.836** | 0.000 | 0.749 | 0.025 |
| 10 | — | — | 0.760 | 0.012 |
| 15 | — | — | 0.761 | 0.006 |
| 25 | — | — | 0.761 | 0.000 |

inability (AUC) while $A_{\mathrm{median}}$ maximizes the raw effect size ($|\gamma|$). In addition, we observe that the signal is robust to the choice of operator. Every operator yields AUC above chance by epoch 2, and the ranking of operators is stable across epochs (with the exception of epoch 0, where signals are near chance). The choice of operator affects magnitude more than direction, confirming that the PSH holds across a broad family of summary statistics.

> **Finding (RQ5).** $A_{\mathrm{mean}}$ is the recommended default (peak AUC 0.777); $A_{\mathrm{median}}$ is the best robust alternative (peak AUC 0.740, highest $|\gamma|$). All operators converge above chance ($> 0.6$) from epoch 3 and maintain a stable ranking thereafter, confirming that the memorization signal is robust to the choice of aggregation function.

### 6.6 RQ6: Impact of Neighborhood Size and Similarity Threshold

**Setup.** Each perturbation in PEARL's neighborhood is assigned to one of five input-similarity buckets, [0.50, 0.60), [0.60, 0.70), [0.70, 0.80), [0.80, 0.90), and [0.90, 1.00]. We study two dimensions on Pythia-1.4B at epoch 10. First, the effect of restricting neighbors to a minimum similarity threshold $\alpha$, and Second, the effect of varying the number of perturbations $K$ per sample.

**Effect of the similarity threshold.** Table 6 reports the memorization gap and AUC when only perturbations above a given input-similarity threshold are retained. We observe that restricting to high-similarity neighbors monotonically *improves* detection quality. The AUC rises from 0.761 (all perturbations, $\alpha \geq 0.5$) to **0.836** when only the closest bucket ($\alpha \geq 0.90$) is used.

**Effect of the number of perturbations.** Table 7 reports AUC as a function of the number of neighbors $K$ per sample, subsampled randomly from available perturbations (30 trials). We report two regimes. First, using only the high-quality [0.90, 1.00] bucket (*targeted*) and second, using all five buckets (*pooled*).

We observe that restricting to high-similarity neighbors improves detection. As shown in Table 7, AUC rises from 0.761 at $\alpha \geq 0.50$ to 0.836 at $\alpha \geq 0.90$, while $|\gamma|$ increases from 0.213 to 0.286 (Table 6). Both $\mu(S_{\mathcal{G}})$

and $\mu(S_\mathcal{E})$ rise as the threshold increases. This divergence is precisely the mechanism predicted by the PSH. That is, at high similarity, generalized instances remain stable while memorized instances show brittleness; at lower similarity, both populations degrade together, suppressing the gap. Coverage remains 100% across all thresholds, confirming that $\alpha = 0.90$ is practically achievable using the surface-level transformation operators. These results constitute the empirical justification for the $\alpha = 0.90$ default used throughout Sections 5.3–6.7. In the targeted setting ($\alpha \geq 0.90$), AUC grows from 0.759 ($K{=}1$) to 0.836 ($K{=}5$) and then saturates. The standard deviation collapses from 0.019 at $K{=}1$ to 0.000 at $K{=}5$, meaning that with five high-similarity neighbors, the score is fully deterministic. In the pooled setting, convergence is slower and even at $K{=}25$ the pooled AUC (0.761) does not match the targeted AUC at $K{=}5$ (0.836), making the targeted configuration strictly superior. In addition, we observe that even a single high-quality perturbation is informative. For instance, a single neighbor from the [0.90, 1.00] bucket yields AUC 0.759. This confirms that the memorization signal is present at the individual perturbation level, with practical implications for low-resource settings where generating many perturbations per instance is costly.

> **Finding (RQ6).** Both the similarity threshold $\alpha$ and the neighborhood size $K$ have a strong, monotone effect on detection quality. Restricting perturbations to input similarity $\geq 0.90$ improves AUC from 0.759 to 0.836 on Pythia-1.4B at epoch 10, while lower-similarity buckets used alone yield near-chance performance (AUC $\leq 0.532$). A neighborhood of $K{=}5$ high-similarity perturbations is sufficient to saturate the signal; variance is negligible at $K{\geq}5$. We recommend $\alpha{=}0.90$ and $K{=}5$ as the default PEARL configuration.

### 6.7 RQ7: Memorization vs. Membership

**Setup.** Membership inference attacks (MIAs) and PEARL answer fundamentally different questions. MIA asks, *was this instance used during training?* while PEARL asks, *does the model behave differently on this instance than on semantically-equivalent perturbations?* Every memorized instance is a member, but not every member is memorized. A model may have seen a data point during training and nonetheless learned to generalize from it. We evaluate this distinction on Pythia-1.4B at epoch 10 (1,000 members, 1,000 non-members). The results are reported in the Table **??** where flags are assigned at the Youden's J optimal threshold for each method independently.

**PEARL and MIAs remain substantially complementary.** Despite PEARL's AUC advantage, Jaccard coefficients range from 0.458 to 0.479: the two methods agree on fewer than half their detections. PEARL exclusively flags 185 members missed by all three MIA methods. MIA methods exclusively flag 202 members missed by PEARL. This complementarity is not eliminated by PEARL's superior AUC; rather, it reflects fundamentally different aspects of the memorization phenomenon. The ensemble of PEARL and any MIA identifies 811 of 1,000 true members (185 + 424 + 202). This coverage exceeds the best individual method (PEARL, 609) by 20% in absolute terms, confirming that the two detection families are complementary auditing tools even when one dominates on AUC.

> **Finding (RQ7).** Together both PEARL and MIA identify 811/1,000 true members, a 20% absolute gain. This indicate that the two families are complementary auditing tools.

### 6.8 RQ8: Task Dependency (Code Synthesis vs Text Completion)

**Setup.** We evaluate PEARL on Pythia-1.4B fine-tuned on competitive programming problems, using 500 member instances drawn from CodeContest (Li et al., 2022) and 500 non-member instances drawn from OpenCodeInstruct (Ahmad et al., 2025). Checkpoints are saved at epoch 1 and epoch 10. The performance measure $m_t$ are both instantiated as embedding cosine similarity, consistent with the text-domain evaluation. All six aggregation operators are evaluated. Tables 8 and 9 report the memorization gap and AUC for all operators at both epochs, alongside MIA baselines.

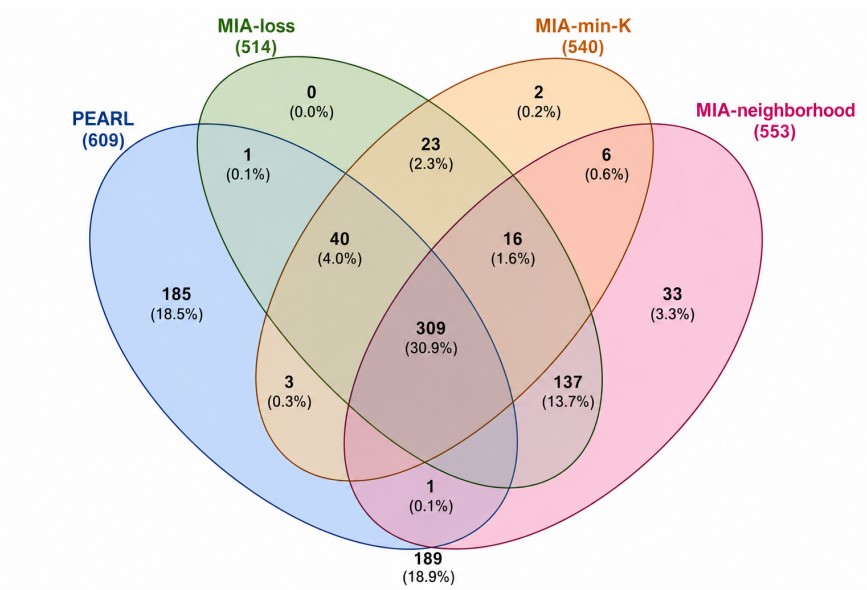

Figure 6: Instance-level overlap between PEARL ($A_{\mathrm{mean}}$, Youden-J threshold) and three MIA methods on Pythia-1.4B at epoch 10 ($N$=1,000 true members). Each number indicates the count of instances lying exclusively in that intersection region. The central region (all four methods, 309) is the largest single region, followed by the PEARL-exclusive region (185, blue) and the MIA-consensus-without-PEARL region (137, lower right). Together the two detection families cover 811/1,000 true members.

Table 8: Memorization gap for code synthesis (Pythia-1.4B, $N_E = N_G = 500$). $|\gamma|$ grows from epoch 1 to epoch 10, confirming progressive memorization consistent with the text domain.

| Epoch | $\mu(S_{\mathcal{G}})$ | $\mu(S_{\mathcal{E}})$ | $|\gamma|$ |
|---|---|---|---|
| 1 | 0.459 | 0.300 | 0.159 |
| 10 | 0.472 | 0.303 | **0.170** |

Table 9: AUC per aggregation operator for PEARL (code synthesis, Pythia-1.4B) and MIA baselines at epochs 1 and 10. **Bold** = best PEARL value per epoch. For reference, text-domain AUC at epoch 10

| Method | Operator / Score | Code (epoch 1) | Code (epoch 10) | Text (epoch 10) |
|---|---|---|---|---|
| PEARL | $A_{\mathrm{mean}}$ | 0.785 | 0.828 | 0.761 |
| | $A_{\mathrm{min}}$ | **0.873** | **0.914** | 0.588 |
| | $A_{\mathrm{median}}$ | 0.844 | 0.870 | 0.698 |
| | $A_{q_{0.25}}$ | 0.864 | 0.906 | 0.644 |
| | $A_{q_{0.10}}$ | 0.868 | 0.910 | 0.622 |
| | $A_{\mathrm{neg\text{-}var}}$ | 0.679 | 0.788 | 0.681 |
| *MIA baselines* | Loss | 0.858 | 0.899 | 0.663 |
| | Min-K% | 0.862 | **0.910** | 0.668 |
| | Neighborhood Attack | 0.870 | 0.896 | 0.664 |

**PEARL detects code memorization substantially better than text memorization.** Every operator achieves higher AUC on code than on text, but gains are not uniform. Lower-tail and worst-case operators improve the most, while the mean improves the least. As shown in Table 10, $A_{\mathrm{min}}$ gains +0.326 (the largest of any operator), $A_{q_{0.10}}$ gains +0.288, $A_{q_{0.25}}$ gains +0.262, while $A_{\mathrm{mean}}$ gains only +0.067 (the smallest). The operators that underperform on text are precisely the ones that excel on code. relies on.

Table 10: Operator ranking by AUC for text (epoch 10, Pythia-1.4B) vs. code (epoch 10, Pythia-1.4B). The complete rank reversal of $A_{\min}$ (6th $\rightarrow$ 1st) and the corresponding inversion of the top and bottom operators confirm that the optimal aggregation strategy must be calibrated to the task's memorization structure. Spread $= \max(\text{AUC}) - \min(\text{AUC})$ across all operators.

| Rank | Text operator | AUC | | Code operator | AUC |
|:---:|:---|:---:|:---:|:---|:---:|
| 1 | $A_{\text{mean}}$ | 0.761 | ‖ | $A_{\min}$ | **0.914** |
| 2 | $A_{\text{median}}$ | 0.698 | ‖ | $A_{q_{0.10}}$ | 0.910 |
| 3 | $A_{\text{neg-var}}$ | 0.681 | ‖ | $A_{q_{0.25}}$ | 0.906 |
| 4 | $A_{q_{0.25}}$ | 0.644 | ‖ | $A_{\text{median}}$ | 0.870 |
| 5 | $A_{q_{0.10}}$ | 0.622 | ‖ | $A_{\text{mean}}$ | 0.828 |
| 6 | $A_{\min}$ | 0.588 | ‖ | $A_{\text{neg-var}}$ | 0.788 |
| Spread ($\Delta_{\text{AUC}}$) | | 0.173 | ‖ | | 0.126 |

At epoch 10, PEARL ($A_{\min}$ 0.914) and the best MIA method (Min-K% 0.910) differ by only $+0.004$ — statistical parity. At epoch 1, the gap is equally small. PEARL ($A_{\min}$ 0.873) vs. MIA-Neighborhood (0.870), a difference of $+0.003$. This convergence reflects the structural properties of code. That is, code has much lower token entropy than natural language, making the log-probability divergence exploited by MIA very strong. Simultaneously, the exact-match nature of verbatim code memorization makes output brittleness equally strong. The two detection principles arrive at the same discrimination boundary through entirely different pathways, confirming that code synthesis memorization is robustly detectable by both signal types.

**Memorization signal profile is task dependent.** Table 9 shows that the operator ranking is entirely different between text and code. The AUC spread across operators is actually *smaller* for code ($\Delta = 0.914 - 0.788 = 0.126$) than for text ($\Delta = 0.761 - 0.588 = 0.173$). The task dependency is therefore not about some operators failing on code (since all six operators are effective) but about which aspect of the memorization signal is most diagnostic for each task's output structure.

> **Finding (RQ8).** Every operator improves on code relative to text, but gains are inversely proportional to text performance. $A_{\min}$ gains the most ($+0.326$, from worst on text to best on code), $A_{\text{mean}}$ the least ($+0.067$). The complete rank reversal of $A_{\min}$ (rank 6 on text $\rightarrow$ rank 1 on code) demonstrates that the optimal aggregation strategy is task-dependent.

## 7 Discussion

### 7.1 Theoretical Conditions for the PSH to Hold

The PSH is grounded in the geometry of the model's output landscape in input space. It holds most reliably when three conditions are jointly satisfied.

**Condition 1: Sharp input-space landscape for memorized instances.** Memorization corresponds to $x$ lying in a *sharp minimum* of the loss landscape. Formally, the Hessian $\nabla_\theta^2 \mathcal{L}(x, \theta)$ has large positive eigenvalues for memorized $x$. A perturbation $x_i$ effectively induces a parameter shift $\delta\theta$, causing a large loss increase $\delta\mathcal{L} \approx \frac{1}{2}\delta\theta^T \nabla_\theta^2 \mathcal{L}\delta\theta$ for memorized points, manifesting as a performance drop in $m_b$. Generalized points lie in *flat minima*, where the eigenvalues are small, and are thus robust to such perturbations.

**Condition 2: Task alignment with the memorized content.** The task $t$ and performance measure $m_t$ must be sensitive to the specific content that was memorized. For the task of next-token prediction evaluated via output cosine similarity, this condition is satisfied whenever the model has memorized the *exact continuation* of a training input. A perturbation disrupts the retrieval of that continuation and $m_t$ directly captures the resulting divergence. However, this condition can fail when the task does not require the memorized content. For example, if $t$ is summarization and $x$ was memorized, the model can still produce

an acceptable summary from a perturbed input without relying on memorizatim, providing a high $S(x)$ for a genuinely memorized instance, a false negative.

**Condition 3: Semantic preservation within the neighborhood.** The neighborhood function $N(x)$ must generate perturbations $x'$ that are *semantically equivalent* to $x$ while being *syntactically distinct.* This is the condition that ensures any observed performance drop is attributable to memorization-induced brittleness rather than to the model simply receiving a meaningfully different input. Operationally, this requires a sufficiently high similarity threshold $\alpha$ in the embedding space of the neighborhood. As shown empirically in Section 6.6, the detection signal is mostly reliable for $\alpha \geq 0.90$. Transformation operators must therefore be designed to produce perturbations that reliably land in the [0.90, 1.00] similarity bucket. Surface-level transformations (`whitespace`, `change_case`, `swap_chars`) satisfy this requirement. They achieve mean input similarity $\geq 0.957$ with 100% coverage (Section 6.2).

## 7.2 Theoretical Failure Cases

$S(u) < \mu(\mathcal{S}_{\mathcal{E}})$ is sufficient evidence for memorization under the three conditions above, but the absence of brittleness does not guarantee generalization. Specifically, three failure modes can cause PEARL to produce false negatives (missed memorized instances) or false positives (incorrectly flagged generalized instances).

**Failure Case 1: Incomplete generalization (false positives).** A model may have learned a pattern that is robust within a narrow but well-defined input manifold. If a perturbation $x' \in N(x)$ falls at the boundary of this learned manifold, technically within the similarity threshold but outside the region where the model's pattern applies, the model's output will diverge even though the instance is not memorized. This is *incomplete generalization* where a local but imperfect representation mimics memorization brittleness under certain perturbations.

The calibration phase mitigates this risk by establishing the baseline distribution $\mathcal{S}_{\mathcal{G}}$ from known non-members drawn from the Mimir benchmark Duan et al. (2024). The detection threshold $\mu(\mathcal{S}_{\mathcal{E}})$ is set relative to this distribution, so that only instances whose neighborhood score is anomalously low relative to the generalization baseline are flagged. Nonetheless, instances at the manifold boundary of a highly specialized generalized pattern may still yield false positives, particularly at small model sizes where representations are less smooth (Section 6.3).

**Failure Case 2: Robust memorization (false negatives).** The PSH assumes that memorization produces brittle output pathways. However, if a model memorizes a higher-order *rule* or *template* such as a grammatical structure, proof strategy, or code idiom rather than a verbatim sequence, that memorized pattern may itself be robust to surface-level perturbations. In this case, $S(u)$ will remain high even for a memorized instance, producing a false negative.

This failure mode is empirically visible in RQ7(Section 6.7). Of the 1,000 true members evaluated on Pythia-1.4B at epoch 10, PEARL correctly flags 720 but misses 270. A subset of these 270 likely corresponds to instances whose memorized content takes the form of a robust template rather than a brittle verbatim pathway that PEARL cannot detect through output stability analysis. This also explains why MIA methods flag 202 members that PEARL misses.Log-probability signals can detect distributional memorization of token-level patterns even when generation behavior remains stable under perturbation. These two failure modes are therefore complementary. That is, MIA methods are more sensitive to distributional template memorization, while PEARL is more sensitive to generative verbatim memorization.

**Failure Case 3: Misaligned task–metric pair (false negatives).** This failure mode is the empirical consequence of Condition 2 failing. If the metric $m_t$ does not capture the specific information pathway that was memorized, for instance, using output cosine similarity to detect memorization of a highly structured format where small perturbations change structure but not semantic content measurably, the sensitivity signal will be weak even for genuinely memorized instances. This failure mode is task-specific and can be mitigated by selecting $m_t$ to be as closely aligned as possible with the type of memorization under

investigation (e.g., token-level edit distance for verbatim code memorization, or ROUGE precision for factual memorization in QA settings).

# 8 Related Works

We review existing research on memorization in large language models, with a focus on detection techniques, the distinction between memorization and membership inference, and the limitations of current black-box and white-box approaches.

**Memorization vs. Generalization.** The relationship between memorization and generalization in LLMs is complex and often difficult to disentangle. While generalization is the intended outcome of training, memorization, particularly of rare or sensitive data is undesireable leading to privacy concerns and misleading model performance. Studies such as (Tirumala et al., 2022) and (Kiyomaru et al., 2024) show that memorization increases with model scale, prompt length, and training dynamics, even in the absence of overfitting. This distinction becomes even more blurred in the presence of data contamination. Jiang et al. (2024) demonstrate that even minor overlaps between training and evaluation data can significantly inflate performance, making it difficult to determine whether a model is truly generalizing or merely recalling memorized content.

To address this ambiguity, several works have proposed several definitions of memorization, including verbatim, approximate, and counterfactual memorization (Feldman & Zhang, 2020a; Feldman, 2020; Ishihara, 2023; Hartmann et al., 2023a; Zhang et al., 2023). Recent work has deepened our understanding of memorization mechanisms. (Speicher et al., 2024) characterized learning phases in memorization, while (Dankers & Titov, 2024) demonstrated that memorization occurs gradually across model layers, with early layers playing a more crucial role. In an effort to formalize the boundary between memorization and generalization, (Wang et al., 2025) introduce the concept of distributional memorization. Their findings show that LLMs tend to rely more on memorization in knowledge-intensive tasks, such as factual question answering, while generalization is more prominent in reasoning-oriented tasks. To reduce memorization, (Lee et al., 2022) and (Kandpal et al., 2022) propose deduplication techniques that significantly reduce memorization and improve training efficiency. (Ippolito et al., 2023) introduce MEMFREE decoding to block verbatim generation, though paraphrased leakage persists. These findings motivate the need for robust detection methods that can reliably identify memorization, especially in black-box settings where training data and model internals are inaccessible.

**Training Data Extraction and Membership Inference.** Early work by (Carlini et al., 2021b) demonstrated that LLMs can memorize and regurgitate verbatim training data, even when such data appears only once. Their black-box extraction method combines candidate generation with membership inference, revealing that memorization is not necessarily tied to overfitting. (Nasr et al., 2023) scale this approach to production models, introducing divergence-based attacks that extract gigabytes of training data. These methods, while powerful, often rely on access to known training data or extensive sampling infrastructure. (Jagielski et al., 2023) explore forgetting dynamics, showing that early-seen examples are more likely to be forgotten. (Meeus et al., 2024) propose copyright traps to audit memorization in models that do not naturally memorize. While membership inference attacks (MIAs) are often used to test for memorization, they are not equivalent. As (Schwarzschild et al., 2024) argue, membership is not memorization. A model may have seen a data point during training without memorizing it in a way that is problematic or reproducible. These limitations emphasize the need for alternative approaches that directly assess memorization behavior, rather than relying solely on membership status.

**Memorization Metrics and Detection Frameworks.** Recent work has proposed formal metrics to quantify memorization. (Carlini et al., 2023) proposed a prompting-based method to measure exact suffix reproduction, showing that memorization scales with model size and data duplication. (Zhang et al., 2023) propose counterfactual memorization, measuring the change in model predictions when specific training examples are removed. (Guo et al., 2024) use memorization signals for AI-generated text detection. (Schwarzschild et al., 2024) define the Adversarial Compression Ratio (ACR), which identifies memorized

content based on the minimal prompt length required to elicit it. While effective, these approaches often require white-box access or known training data, limiting their applicability in real-world black-box settings. Our work builds on this line by introducing PEARL, a black-box framework that detects memorization via perturbation sensitivity.

## 9    Conclusion

In this paper, we introduce PEARL, a black-box framework that operationalizes the input perturbation sensitivity hypothesis (PSH) for detecting memorization in LLMs. Through extensive experiments on open-source models across diverse situations, we demonstrated that perturbation sensitivity serves as a reliable indicator of memorization, particularly in tasks requiring exact content reproduction. By providing a practical tool for auditing memorization without requiring access to training data or model internals, PEARL contributes to addressing critical challenges in data privacy, model evaluation, and responsible AI development.

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
