# OpenReview forum: "Detecting LLM Memorization through Input Perturbation Analysis"
_TMLR — Under review for TMLR_

### Review · Reviewer_Lgsr · 2026-06-21

**Summary Of Contributions:**

This paper proposes PEARL, a black-box framework for detecting instance-level memorization in LLMs without requiring access to training data, model weights, or logits. The key idea is that memorized instances are more sensitive to semantically preserving input perturbations than generalized instances. Extensive experiments on the Pythia model suite demonstrate that PEARL outperforms several existing black-box and gray-box baselines for memorization detection.

weakness:
- Experiments are limited to relatively small open-source models (up to Pythia-2.8B), leaving questions about scalability to modern frontier LLMs.
- The central hypothesis (memorized instances are more perturbation-sensitive) is primarily empirically validated

**Audience:**

Yes

**Audience Explanation:**

The topic is relevant to the TMLR community, and the proposed black-box approach to memorization detection is practically meaningful. While some of the paper's claims would benefit from stronger validation, the findings are still likely to be of interest to researchers studying LLM memorization, privacy, and auditing.

**Broader Impact Concerns:**

The method could potentially be misused to identify highly memorized training instances and facilitate privacy attacks. However, its primary purpose is privacy auditing and model evaluation, and the overall societal risk appears limited.

**Claims And Evidence:**

No

**Claims Explanation:**

The central claim that perturbation sensitivity is a reliable signature of memorization is less fully supported, because many experiments use membership labels as a proxy for memorization, even though membership and memorization are not equivalent. Overall, the evidence is suggestive and fairly extensive, but not fully convincing for the strongest claims.

**Requested Changes:**

Strictly speaking, the authors demonstrate that: Training-set members tend to exhibit higher perturbation sensitivity than non-members. It is better to clarify the relationship between perturbation sensitivity, membership, and memorization, since the current experiments largely use membership labels as a proxy for memorization.

justify why perturbation sensitivity should be considered a reliable signature of memorization, either through theoretical analysis or more direct empirical validation on genuinely memorized versus generalized instances.

evaluate the method on larger and more recent LLMs to demonstrate scalability beyond the Pythia model family.

---

### Review · Reviewer_418L · 2026-07-18

**Summary Of Contributions:**

This paper introduces PEARL (PErturbation Analysis for Revealing Language-model memorization), a behavioral method for detecting instance-level memorization without requiring access to model logits or parameters. The central hypothesis, termed the Input Perturbation Sensitivity Hypothesis (PSH), is that a memorized input-output association is unusually brittle: small, purportedly semantics-preserving changes to the input cause the model’s output to diverge substantially from the reference continuation. In contrast, examples supported by a generalized representation should exhibit relatively stable outputs in a local perturbation neighborhood.

PEARL generates several perturbations of a candidate input, measures the similarity between the generated continuations and a reference continuation, aggregates these similarities, and compares the resulting score against distributions obtained from known members and non-members. The paper evaluates several perturbation operators, aggregation functions, neighborhood sizes, similarity thresholds, training epochs, model sizes, and two task settings. The main experiments use Pythia models ranging from 70M to 2.8B parameters. The authors report that PEARL performs poorly or only slightly above chance on smaller models but becomes substantially more discriminative at 1.4B and 2.8B parameters. They also show that PEARL and conventional membership-inference methods detect partially non-overlapping subsets of training examples.

The paper has several strengths. The basic behavioral signal is intuitive, the procedure is simple to implement in a black-box setting, and the ablations over perturbations, aggregation functions, neighborhood sample size K, model scale, and repeated exposure are useful. The observation that perturbation sensitivity and log-probability-based membership signals identify complementary examples is also potentially valuable. In my view, PEARL is best understood as one component of a multi-signal memorization audit rather than as a standalone or definitive detector.

However, several central claims are not yet adequately supported. Most importantly, the description of PEARL as “training-data-free” is inconsistent with its calibration procedure, which requires known training members and known non-members from the target distribution. In addition. Finally, the parameter-space Hessian explanation in Section 7 does not directly characterize sensitivity to input perturbations.

Overall, I find the proposed signal interesting, but the current evidence supports a narrower claim than the one made in the paper

**Audience:**

Yes

**Audience Explanation:**

The paper addresses a relevant problem for researchers working on privacy, training-data auditing, contamination detection, robustness, and language-model evaluation. A behavioral detector that operates without token-level logits or model parameters would be useful in black-box settings.

The most interesting empirical result is that local perturbation sensitivity becomes strongly discriminative after repeated fine-tuning for sufficiently large models, while it remains weak for smaller models and early checkpoints. This conditional result is useful even if it limits the method’s scope.

The partial non-overlap between PEARL and standard membership-inference methods is also potentially informative. Although the current experiments do not establish that these methods detect distinct forms of memorization, the different error patterns motivate the development of multi-signal auditing methods.

A controlled evaluation across model families, exposure frequencies, memorization types, and domains could make the contribution substantially more useful. It could provide actionable guidance about when perturbation sensitivity is reliable, which perturbations are appropriate for each domain, and which forms of memorization PEARL systematically misses.

**Claims And Evidence:**

No

**Claims Explanation:**

The experiments provide convincing evidence for a narrower claim: within the studied Pythia models and experimental setting, input-perturbation sensitivity can be a useful signal for detecting some training members, especially at larger model scales. The decay-curve observation is clear, and PEARL performs competitively against the evaluated black-box and gray-box baselines for the larger models. The complementarity result in Figure 6 is also valuable because it suggests that PEARL is not simply reproducing the ranking induced by existing membership-inference scores.

However, several broader interpretations are not yet sufficiently supported.

First, the results show a strong dependence on model scale. PEARL underperforms some existing approaches for models below approximately one billion parameters, and its performance at 1.4B and 2.8B parameters appears only slightly above chance. This raises an important question about the conditions under which PSH is expected to hold. The paper currently discusses model size, but the relevant variable may instead be the relationship between model capacity and fine-tuing dataset's training-token count (training set size vs. model size). Without controlling for or analyzing these factors, it is difficult to determine whether larger parameter count itself causes the stronger perturbation-sensitivity signal.

Second, most of the empirical evidence is based on a single model family. Evidence from another model family or training pipeline is needed to distinguish a general memorization phenomenon from a Pythia-specific result.

Third, the terminology surrounding training-data access requires clarification. Section 4.1 appears to require known members for calibration. A method that does not require access to the target model’s complete training corpus can reasonably be described as not requiring full training-data access, but “training-data-free” would be misleading if labeled member examples are necessary for calibration. The paper should clearly distinguish among no access to the full training set, access to a small labeled calibration set, and completely unsupervised membership detection.

Finally, the theoretical analysis does not yet provide a sufficiently direct explanation for PSH. Section 7 studies a parameter-space Hessian, whereas PEARL is explicitly based on local changes in input space. An analysis based on input Jacobians, token-level sensitivity, hidden-state geometry, or output-distribution curvature with respect to input representations would be more directly aligned with the proposed mechanism. The current parameter-space analysis may be relevant, but the connection between that analysis and the perturbation-based score should be made more explicit.

Overall, the evidence supports PEARL as a promising empirical signal in the evaluated setting. It does not yet convincingly support a general claim that memorized examples are systematically more brittle under small input perturbations across model families, scales, and memorization regimes.

**Requested Changes:**

See above

---

### Review · Reviewer_96wj · 2026-07-19

**Summary Of Contributions:**

This paper proposes PEARL, a black-box method for detecting possible memorization in language models. The main idea is that a memorized example may be more sensitive to small input changes than an example that the model has generalized from. PEARL creates several perturbed versions of an input, compares the resulting model outputs with the reference continuation, and uses the changes in output similarity as a memorization score. They evaluate PEARL on several Pythia models, ranging from 70M to 2.8B parameters. The experiments study model size, training epochs, perturbation types, similarity thresholds, neighborhood size, and aggregation functions. The authors also compare PEARL with several membership inference methods and include an additional experiment on code generation.

I think the paper has several strengths. The problem is important, especially for auditing models that are only available through an API. The basic idea is simple and intuitive, and the paper includes more analysis than a standard method paper. In particular, the results across model sizes and training epochs are useful, and the study of different perturbations helps explain when the method works.

My main concern is that the experiments use training membership as the ground-truth label, rather than memorization itself. The paper correctly argues that membership and memorization are different, but the evaluation still measures how well PEARL separates members from non-members. It is therefore not yet clear whether PEARL is detecting memorization, training-set familiarity, or another form of local instability.

I also think the claim that the method requires no training-data access should be stated more carefully. The detection procedure does not require searching the full training corpus, but the calibration step still requires known members and known non-members. There are also some concerns about the code experiment, semantic preservation of the perturbations, and several inconsistent numbers in the paper.

**Audience:**

Yes

**Audience Explanation:**

I think the topic would be interesting to part of the TMLR audience. Memorization detection is closely related to privacy, copyright, benchmark contamination, and model auditing. A method that only needs input-output access would be useful because many current language models are available only through black-box APIs.

**Broader Impact Concerns:**

PEARL should not be presented as definitive proof that a specific document was included in a model’s training data. The method may produce false positives for unusual, poorly tokenized, or out-of-distribution inputs. It may also miss memorized examples that remain stable under the selected perturbations.

**Claims And Evidence:**

No

**Claims Explanation:**

The experiments show that training members and non-members have different sensitivity to the perturbations used by PEARL. I find this result interesting. However, I do not think the current evidence fully supports the paper’s broader claim that PEARL detects memorization.

The main issue is the evaluation label. The positive examples are training members, but the paper itself notes that being a member of the training set does not necessarily mean that the example was memorized. A model may have seen an example and still learned a general pattern from it. Therefore, an AUC computed using member/non-member labels is mainly an evaluation of membership separation. It is not a direct evaluation of memorization detection.

This also affects the comparison with membership inference. The paper finds that PEARL and MIA methods flag different examples, which is a useful observation. However, disagreement between two methods does not by itself show that one method detects verbatim memorization while the other detects distributional or template memorization. The disagreement could also come from different errors, thresholds, or sensitivities. The PEARL-only and MIA-only examples need further analysis before these interpretations can be made.

The code-generation experiment has another issue. The members come from CodeContest, while the non-members come from OpenCodeInstruct. Since these are different datasets, the high AUC may partly reflect differences in style, formatting, problem type, or data source. A held-out non-member set from the same dataset would provide a cleaner experiment.

Finally, semantic preservation is mainly determined using embedding cosine similarity. This is a reasonable first step, but it is not enough to show that all perturbations preserve meaning. Character swaps may create spelling errors or change tokenization. In code, whitespace, casing, and character changes may even alter syntax or program behavior. Some of the measured sensitivity may therefore come from malformed or unusually tokenized inputs rather than memorization.

**Requested Changes:**

(1) The paper needs a more direct evaluation of memorization. The current experiments use membership labels, which do not provide ground truth for memorization. The authors should add an experiment in which memorization is known or can be independently measured. Controlled canaries, examples with different repetition counts, exact continuation extraction, or models trained with and without selected examples would all be reasonable options.

(2) The authors should also discuss how these calibration examples would be obtained in a realistic black-box setting. An experiment with limited, noisy, or distribution-shifted calibration data would help show how practical the method is.

(3) Calibration and final evaluation should use separate data. It is not clear whether the Youden threshold and other choices are selected using the same data used for final reporting. The authors should use separate calibration, validation, and test sets. Perturbation type, aggregation rule, similarity threshold, neighborhood size, and model checkpoint should be selected without using the final test labels.

(4) The code experiment should use members and non-members from the same source distribution. Using CodeContest for members and OpenCodeInstruct for non-members introduces a strong dataset confound. A cleaner design would train on one part of CodeContest and use a held-out part of CodeContest as non-members. The examples should also be matched by length, topic, and formatting, where possible.

(5) Semantic preservation needs stronger validation. The authors should manually inspect and label a sample of the perturbations, or use task-specific validity checks. For code, the paper should report whether the modified inputs remain syntactically valid and preserve the original problem meaning. It would also be useful to study whether the score is driven by changes in tokenization. For example, the authors could compare perturbations with similar embedding similarity but different token-edit distances.

(6) The interpretation of the PEARL-only and MIA-only examples should be supported by evidence. The current overlap analysis shows that the methods are different, but it does not explain why. The authors should compare the PEARL-only, MIA-only, both, and neither groups using an independent measure such as exact-match generation, extraction success, continuation likelihood, or counterfactual influence.